# Infections during Non-Neutropenic Episodes in Pediatric Cancer Patients—Results from a Prospective Study in Two Major Large European Cancer Centers

**DOI:** 10.3390/antibiotics11070900

**Published:** 2022-07-05

**Authors:** Stefan Schöning, Anke Barnbrock, Konrad Bochennek, Kathrin Gordon, Andreas H. Groll, Thomas Lehrnbecher

**Affiliations:** 1Pediatric Hematology and Oncology, Hospital for Children and Adolescents, Johann Wolfgang Goethe-University, 60590 Frankfurt, Germany; stefan.schoening@kgu.de (S.S.); anke.barnbrock@kgu.de (A.B.); konrad.bochennek@kgu.de (K.B.); 2Infectious Disease Research Program, Department of Pediatric Hematology and Oncology and Center for Bone Marrow Transplantation, University Children’s Hospital Münster, 48149 Münster, Germany; kathrin.gordon@ukmuenster.de (K.G.); andreas.groll@ukmuenster.de (A.H.G.)

**Keywords:** child, cancer, infection, neutropenia, non-neutropenic episode

## Abstract

Whereas the clinical approach in pediatric cancer patients with febrile neutropenia is well established, data on non-neutropenic infectious episodes are limited. We therefore prospectively collected over a period of 4 years of data on all infectious complications in children treated for acute lymphoblastic or myeloid leukemia (ALL or AML) and non-Hodgkin lymphoma (NHL) at two major pediatric cancer centers. Infections were categorized as fever of unknown origin (FUO), and microbiologically or clinically documented infections. A total of 210 patients (median age 6 years; 142 ALL, 23 AML, 38 NHL, 7 leukemia relapse) experienced a total of 776 infectious episodes (571 during neutropenia, 205 without neutropenia). The distribution of FUO, microbiologically and clinically documented infections, did not significantly differ between neutropenic and non-neutropenic episodes. In contrast to neutropenic patients, corticosteroids did not have an impact on the infectious risk in non-neutropenic children. All but one bloodstream infection in non-neutropenic patients were due to *Gram*-positive pathogens. Three patients died in the context of non-neutropenic infectious episodes (mortality 1.4%). Our results well help to inform clinical practice guidelines in pediatric non-neutropenic cancer patients presenting with fever, in their attempt to safely restrict broad-spectrum antibiotics and improve the quality of life by decreasing hospitalization.

## 1. Introduction

Infections still represent a major cause for morbidity and mortality in pediatric cancer patients and affect the quality of life during treatment. Neutropenia is a well-established risk factor for bacterial and fungal infections [1], and the prompt initiation of intravenous antibiotics that cover both *Gram*-positive and *Gram*-negative pathogens dramatically decreased mortality and has become the standard of care in neutropenic children with cancer [2]. For this setting, pediatric specific guidelines, which address both indication and options for prophylaxis, empirical therapy and treatment of documented infections have been developed over the last few years and are regularly updated [3,4,5]. In contrast, data on infections occurring in non-neutropenic pediatric cancer patients are still limited, and despite validated risk-prediction models for bloodstream infections [6,7,8], the clinical approach in this setting varies widely across different institutions ranging from no administration of antibiotics to intravenous combination therapy [9]. In order to add data for the development of clinical practice guidelines in this setting, we analyzed all infectious episodes, categorized into fever of unknown origin (FUO), microbiologically documented and clinically documented infections that occurred in non-neutropenic periods in children with cancer treated in two major pediatric cancer centers over a period of four years.

## 2. Results

A total of 210 patients [145 boys (69%), 65 girls (31%)] were included in the analysis. The median age was 6 years (range, 2 months to 17 years), with 117 patients (55.7%) ≤ 6 years, 55 (26.2%) between 6 and 12 years, and 38 (18.1%) older than 12 years. Underlying diagnoses were as follows: ALL 142 patients [(67.6%; standard risk (SR) 61 (29.0%), intermediate risk (MR) 46 (21.9%), and high risk (HR) 35 (16.7%)], AML 23 (10.9%), relapse of ALL 5 and AML 2 (2.4% and 1%, respectively) and NHL 38 (18.7%). The patients received a total of 621 chemotherapy cycles.

For the analysis of infectious episodes occurring in the different patient populations, we excluded the seven patients with leukemia relapse due to the small number and the highly variable treatment. This resulted in the analysis of a total of 776 infectious episodes observed in 203 patients, with 571 infections occurring during neutropenia (73.6%) and 205 infections occurring during a period without neutropenia (26.4%), respectively. The proportion of infections during periods without neutropenia was similar for all underlying malignancies [ALL 152 (28.1%), AML 25 (21.9%), and NHL 28 (23.1%)]. The distribution of FUO, microbiologically documented and clinically documented infections did not significantly differ between neutropenic and non-neutropenic episodes, with 65.2% and 66.3% for FUO, 26.1% and 23.4% for microbiologically documented infections, and 8.7% and 10.3% for clinically documented infections, respectively. Similar results were also seen for the subgroups of patients with ALL, AML, and NHL (Table 1). Out of the 205 infectious episodes without neutropenia, 105 (51.2%) occurred in patients who had received corticosteroids, whereas in 368 (64.4%) of the 571 neutropenic infectious episodes, corticosteroids had been administered (*p* = 0.001). The proportion of infections without neutropenia, in whom the patient had received corticosteroids, was 71.4% in NHL, 55.9% in ALL patients, and 0% in patients with AML (Table 1).

Of all infections, there was a significantly higher percentage of bloodstream infections in neutropenic than in non-neutropenic patients [86 (15.1%) versus 11 (5.4%); *p* = 0.0003], which was observed in all patient populations [14.4% versus 6.6% for ALL (*p* = 0.01), 20.2% versus 4.0% for AML [not significant (ns)], and 12.9% versus 0% for NHL (ns)]. The use of corticosteroids did not significantly impact the risk of bloodstream infection in non-neutropenic patients (6 patients out of 105 with corticosteroids, 5 patients out of 100 without corticosteroids). All but one bloodstream infection in non-neutropenic patients was due to a *Gram*-positive pathogen, whereas in neutropenic patients, *Gram*-positive bacteremia were seen twice as often as *Gram*-negative bacteremia (ns) (Table 2). In neutropenic patients, bacteremia were caused by a variety of pathogens including staphylococci, streptococci, enterococci, *Klebsiella* spp., *P. aeruginosa*, and *Acinetobacter* spp., whereas all *Gram*-positive bacteremia in non-neutropenic patients were due to staphylococci (in the vast majority coagulase-negative staphylococci) and *Micrococcus* spp. Polymicrobial infections were seen only in neutropenic patients. All children with bacteremia occurring in a non-neutropenic period received antibacterial treatment as inpatient, and survived after an uneventful clinical course. One patient was transferred to the intensive care unit for a reason unrelated to the infection (adverse reaction to pain medication).

Viral infections were diagnosed in 49 (8.5%) of the infections during neutropenia and in 22 (10.9%) of the infection during non-neutropenic periods (Table 2). No significant differences in the distribution of pathogens were seen between neutropenic and non-neutropenic patients and across the underlying diagnoses (Table 2).

Clinically documented infections without a pathogen isolated were seen in a total of 50 (8.7%) neutropenic and in 20 (10.3%) of non-neutropenic episodes. These infections mostly affected the lung, the gastrointestinal tract and the soft tissues (Table 2). No significant differences in the distribution of clinically documented infections were seen between neutropenic and non-neutropenic patients (Table 2), across the underlying diagnoses and between patients with or without corticosteroids (data not shown).

Nine of the patients with an infection occurring during a non-neutropenic period required oxygen support, and one patient non-invasive ventilation. A total of four patients were transferred to the intensive care unit, one due to a non-infectious complication (adverse reaction to pain medication), two due to respiratory problems due to metapneumovirus and pulmonary aspergillosis, respectively, and one due to sepsis syndrome without a pathogen identified. The latter three patients, all diagnosed with ALL and receiving induction therapy, required mechanical ventilation (2 patients) and extracorporeal membrane oxygenation (ECMO) (1 patient). All three died due to the infectious complication, resulting in an infection-related mortality of 1.4% during non-neutropenic episodes.

## 3. Discussion

In contrast to well established guidelines in pediatric cancer patients presenting with febrile neutropenia [3,4,5], data on infections with onset during a non-neutropenic period are limited in children with cancer [8,10,11]. Therefore, it is not surprising that the clinical approach in this setting varies widely across institutions [9], and the use of fluoroquinolones is controversial [4,8]. Our study in 203 patients diagnosed with de novo ALL, AML or NHL showed that almost three quarters of infectious complications occurred during neutropenia, whereas only one quarter was diagnosed during periods without neutropenia. This is in line with the data in 91 pediatric patients with acute leukemia in whom 32.7% of infections were not associated with neutropenia [12] or with the results of an analysis in 108 children (73 diagnosed with leukemia/lymphoma) in whom 24.8% of bacteremia occurred in non-neutropenic patients [13]. For all underlying malignancies, ALL, AML and NHL, the distribution of infectious complications, namely FUO, microbiologically documented and clinically documented infections did not significantly differ between neutropenic and neutropenic infections, which is similar to previous results [12]. A recent study in children with AML demonstrated that the duration of corticosteroid exposure was independently associated with bacteremia and microbiologically documented sterile site infection [14]. This observation corroborates our results in neutropenic patients, where we found twice as many microbiologically documented infections in patients receiving corticosteroids compared to those who did not. In contrast, we did not find that the use of corticosteroids increased the risk for FUO, microbiologically and clinically documented infections, respectively, in non-neutropenic children, which has not been reported before and might be explained by the additive immunosuppressive effects of neutropenia and exposure to corticosteroids.

We observed a significantly higher percentage of bloodstream infections in neutropenic compared to non-neutropenic patients (15.1% versus 5.4%), which was seen in all subgroups of patients, namely those with ALL, AML and NHL. Similar results have been described in a study in pediatric patients with leukemia and solid tumors, which reported on 20% versus 3% of bacteremia in neutropenic and non-neutropenic episodes, respectively [15]. In our analysis, all but one bloodstream infection in non-neutropenic patients were due to a *Gram*-positive pathogen, whereas previous data report on a relevant percentage of *Gram*-negative pathogens also in the non-neutropenic setting [10,11,12], which might be related to the presence of mucositis and barrier damage of the gut in these patients. Whether biomarkers such as procalcitonin are reliable in distinguishing *Gram*-positive from *Gram*-negative infections in pediatric cancer patients need to be evaluated in large clinical trials [16]. In contrast to our approach, others have categorized patients with non-neutropenic fever and central venous line in high-risk and low-risk patients [17]. The latter consisted of well-appearing patients without a focus of infection. Importantly, compared to high-risk patients, the risk for bacteremia was considerably lower in the low-risk setting (8% versus 1.6%), and none of the blood isolates of low-risk patients was a *Gram*-negative pathogen. Whereas one study found a positive blood culture for *Candida* spp. in 32 out of 6140 pediatric and adult cancer patients [18], *Candida* spp. was not isolated in any blood culture of the present analysis.

In our analysis of 205 non-neutropenic infectious episodes, four patients were transferred to the ICU, and three of them died due to either invasive aspergillosis, metapneumovirus or sepsis syndrome without a pathogen identified, respectively. Using an approach with the administration of empirical intravenous antibiotic therapy in both febrile neutropenic and non-neutropenic patients, the infection-related mortality of 1.4% in our analysis was lower than that in a previous study, in which seven out of 188 patients died of gram-negative sepsis, resulting in a mortality of 4% [15]. On the other hand, others did not observe any mortality in well-appearing non-neutropenic children with an infectious episode [11], which underlines the usefulness of risk-stratification in this setting.

Our analysis is limited by the fact that we did not test a predefined clinical approach in non-neutropenic cancer patients with suspected infection. We also recognize that our results may not be generalizable as they were not collected in a multicenter study. However, our data demonstrate that the risk of bloodstream infections in the presence of indwelling central venous catheters is significantly lower in non-neutropenic compared to neutropenic patients. In addition, in non-neutropenic patients, the risk for bloodstream infections is not affected by the use of corticosteroids, which has an important implication for the daily clinical setting. A severe clinical course was rare, and viral and fungal pathogens were responsible for two of the three lethal events. Our results will help to inform clinical practice guidelines in pediatric non-neutropenic cancer patients presenting with fever, in their intention to safely restrict broad-spectrum antibiotics and to improve the quality of life by decreasing hospitalization.

## 4. Patients and Methods

### 4.1. Patients

All children and adolescents up to 18 years of age who were diagnosed between 1 April 2014 and 31 March 2018, and were treated at the University Children’s Hospitals of Frankfurt or Münster, Germany, for de novo acute lymphoblastic leukemia (ALL), acute myeloid leukemia (AML), relapse of acute leukemia, or non-Hodgkin lymphoma (NHL) were enrolled in this prospective observational study (DRKS00006341). Patients were treated according to Berlin-Frankfurt-Münster (BFM)-based protocols (e.g., AIEOP-BFM ALL 2009, AML-BFM 2012 registry, ALL-REZ BFM registry, or B-NHL 2013). Patients could be enrolled twice in the study (e.g., if they received treatments for both de novo and relapsed leukemia). The study was approved by the local Ethical committees of Frankfurt (348/13) and Münster (2014-048-b-S). All patients and/or caregivers have signed informed consent.

### 4.2. Supportive Care

Antibacterial prophylaxis was not routinely administered, whereas patients at high risk for invasive fungal disease (IFD), e.g., patients with AML or recurrent leukemia, received mold-active antifungal prophylaxis [19]. According to current pediatric guidelines, both centers routinely performed diagnostics such as blood cultures in febrile patients or imaging studies in children at highest risk for IFD, e.g., those with persistent febrile neutropenia not responding to broad-spectrum antibiotics after 96 h [3,19]. In the presence of an indwelling central venous catheter, which was present in all patients, intravenous empirical antimicrobial therapy was administered irrespective of the neutrophil count to patients with fever or suspected infection according to pediatric specific guidelines [3,20].

### 4.3. Data Collection and Definitions

An electronic database (secuTrial^®^) was used for data collection, which included demographic data, disease characteristics, and data on laboratory and other diagnostic studies. Data were collected for each cycle of intensive chemotherapy, defined by the first day of chemotherapy until the start of the next cycle of chemotherapy. Fever was defined as temperature higher than 38.5 °C once or between 38.0 °C and 38.5 °C twice within a 4-h interval, and neutropenia as an absolute neutrophil count ≤ 500/mm^3^ at or within three days after the onset of fever [21]. Infections were categorized as a fever of unknown origin (FUO), as microbiologically documented or as a clinically documented infection. Bacteremia was defined as a fever with at least a single positive blood culture for bacteria isolated from peripheral blood or from the central venous indwelling catheter, which also could be a potential skin contaminant such as coagulase-negative staphylococci in children with an intravascular catheter [21]. The association of clinical symptoms with a pathogen (such as *Clostridioides difficile*) was required for the diagnosis of an infection of the gastrointestinal tract, whereas diarrhea alone or the recovery of a pathogen in the stool without clinical symptoms was not sufficient to diagnose gastrointestinal tract infection [21,22]. Pneumonia was diagnosed by a pathological chest X-ray and/or computed tomography scan accompanied by clinical symptoms of lower respiratory infection [21]. Invasive fungal disease was defined according to the revised definitions of the EORTC/MSC (European Organization for Research and Treatment of Cancer/Invasive Fungal Infections Cooperative Group and the National Institute of Allergy and Infectious Diseases Mycoses Study Group) consensus group [23]. The use of corticosteroids was defined by the systemic administration of a dosage of 20 mg/m^2^ prednisone for at least five days during the last three weeks prior to the infectious episode. Corticosteroids, such as dexamethasone, were converted to prednisone equivalents by multiplying the dose of corticosteroid administered by the equivalent pharmacologic dose (for example, 6.7 for dexamethasone) [14].

### 4.4. Statistical Analysis

GraphPad Prism (GraphPad Prism Software version 5.04 for Windows, GraphPad Software, La Jolla, CA, USA) was used for the analysis of the data. Groups were compared by the χ^2^ test or Fisher’s exact test, when applicable. A two tailed *p*-value ≤ 0.05 was considered to be statistically significant.

## Figures and Tables

**Table 1 antibiotics-11-00900-t001:** Fever of unknown origin, microbiologically documented and clinically documented infections in children with acute lymphoblastic leukemia, acute myeloid leukemia, and non-Hodgkin lymphoma.

			Total	ALL				AML	NHL
				ALL-Total (*n* (% *))	ALL-SR	ALL-MR	ALL-HR		
FUO	Neutropenia		372 (65.2)	247 (63.5)	80 (61.1)	49 (59.0)	118 (67.4)	55 (61.8)	70 (75.3)
		with steroids	238	179	51	32	96	3	56
		w/o steroids	134	68	29	17	22	52	14
	No Neutropenia		136 (66.3)	96 (63.2)	37 (69.8)	33 (66.0)	26 (53.1)	19 (76.0)	21 (75)
		with steroids	68	51	17	17	17	0	17
		w/o steroids	68	45	20	16	9	19	4
Microbiologically documented	Neutropenia		149 (26.1)	105 (27.0)	31 (23.7)	22 (26.5)	52 (29.7)	29 (32.6)	15 (16.1)
		with steroids	98	82	25	13	44	2	14
		w/o steroids	51	23	6	9	8	27	1
	No Neutropenia		48 (23.4)	40 (26.3)	11 (20.8)	11 (22.0)	18 (36.7)	5 (20.0)	3 (10.7)
		with steroids	27	26	3	6	17	0	1
		w/o steroids	21	14	8	5	1	5	2
Clinically documented	Neutropenia		50 (8.7)	37 (9.5)	20 (15.2)	12 (14.5)	5 (2.9)	5 (5.6)	8 (8.6)
		with steroids	32	25	14	6	5	0	7
		w/o steroids	18	12	6	6	0	5	1
	No Neutropenia		21 (10.3)	16 (10.5)	5 (9.4)	6 (12.0)	5 (10.2)	1 (4.0)	4 (14.3)
		with steroids	10	8	3	3	2	0	2
		w/o steroids	11	8	2	3	3	1	2

FUO fever of unknown origin, ALL: acute lymphoblastic leukemia, AML: acute myeloid leukemia, NHL: non-Hodgkin lymphoma; SR: standard risk, MR: intermediate risk, HR: high risk; w/o without. * percent: number in relation to neutropenic infections or non-neutropenic infections, respectively.

**Table 2 antibiotics-11-00900-t002:** Bacteremia, viral infections and clinically documented infections in neutropenic and non-neutropenic cancer patients.

Bacteremia					
Non-Neutropenic			Neutropenic		
	***Gram*+**	**10 (91%)**		***Gram*+**	**53 (62%)**
	*Staphylococcus* spp.	1		*Staphylococcus* (CoNeg)	19
	*Staphylococcus* (CoNeg)	6		*S. aureus*	1
	*Micrococccus* spp.	3		*Streptococcus* (viridans group)	15
				*Enterococcus* spp.	4
				*Micrococccus* spp.	6
				Other	8
	***Gram*−**	**1 (9%)**		***Gram*−**	**25 (29%)**
	*P. aeruginosa*	1		*Klebsiella* spp.	2
				*P. aeruginosa*	6
				*Eschericia coli*	13
				*Acinetobacter* spp.	1
				Other	3
				Polymicrobial (*Gram*+ and *Gram*−)	8 (9%)
**Viral infection**					
**Non-neutropenic ***		**22**	**Neutropenic ***		**49**
	Influenza virus	6		Influenza virus	7
	Parainfluenza virus	2		Parainfluenza virus	13
	Herpes simplex virus	3		Adenovirus	1
	Varicella Zoster virus	1		Herpes simplex virus	7
	Enterovirus	4		Enterovirus	10
	RSV	7		RSV	13
				Metapneumovirus	2
**Clinically documented infection**					
**Non-neutropenic**		**20**	**Neutropenic**		**50**
	Lung	14		Lung	29
	Gastrointestinal tract	1		Gastrointestinal tract	5
	Soft tissue	2		Soft tissue	11
	Others **	3		Others **	5

* infections may be due to several viruses; ** others include skin, urinary tract or bone infections.

## Data Availability

By the corresponding author by request.

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
