# Peer review of "Infections during Non-Neutropenic Episodes in Pediatric Cancer Patients—Results from a Prospective Study in Two Major Large European Cancer Centers"

_antibiotics, 2022, doi:10.3390/antibiotics11070900_

Round 1
Reviewer 1 Report
In this manuscript is presented the results from a Prospective Study in Two Major Large European Cancer Centers related to Infections During Non-Neutropenic Episodes in Pediatric Cancer Patients. The topic is relevant and the study design and the study procedure are very clear. The article has a clear language and the aim of the study it is clear and interesting. I have, only, minor suggestions for revision:
Table 1- Generally, the title of the table is placed above the table and the spelling of the abbreviations used below the table, such as a footnote. I recommend that the authors change this.
Line 82, line 87-“AML (ns), and 12.9% versus 0% for NHL (ns)]”; “Gram-negative bacteremia (ns)”- It may not be clear to readers who reading the article what the authors mean by "ns". Authors will need to clarify this aspect.
Line 113-ECMO-please spell the abbreviation
Author Response
Replies to reviewer #1
Comments and Suggestions for Authors
In this manuscript is presented the results from a Prospective Study in Two Major Large European Cancer Centers related to Infections During Non-Neutropenic Episodes in Pediatric Cancer Patients. The topic is relevant and the study design and the study procedure are very clear. The article has a clear language and the aim of the study it is clear and interesting. I have, only, minor suggestions for revision:
We thank the reviewer for the positive comments
Table 1- Generally, the title of the table is placed above the table and the spelling of the abbreviations used below the table, such as a footnote. I recommend that the authors change this.
According to the reviewer´s suggestion, we have placed the title above the table and the explanations of abbreviations below the table (lines 108/109 and 110-112). We also moved the title of Table 2 (lines 153/154 ).
Line 82, line 87-“AML (ns), and 12.9% versus 0% for NHL (ns)]”; “Gram-negative bacteremia (ns)”- It may not be clear to readers who reading the article what the authors mean by "ns". Authors will need to clarify this aspect.
According to the reviewer´s suggestion, we clarified the apprevition “ns” (line 83)
Line 113-ECMO-please spell the abbreviation
According to the reviewer´s suggestion, we have spelled out ECMO (line 120 )
Reviewer 2 Report
The manuscript "Infections During Non-Neutropenic Episodes in Pediatric 2 Cancer Patients – Results from a Prospective Study in Two 3 Major Large European Cancer Centers" provides valuable information to the clinicians about infections occurring in the pediatric ALL/AML/NHL patient population.
The manuscript is well written, and the text/ tables are easy to follow.
The discussion, including the limitations of the study is OK, but a few other studies/ references could be added to them, such as:
1. Chen S, Liu S, Yuan X, Wang H, Wen F. Evaluation of Inflammatory Biomarkers in Pediatric Hematology-Oncology Patients With Bloodstream Infection. J Pediatr Hematol Oncol. 2021 May 1;43(4):e596-e600.
2. Chen S, Liu S, Yuan X, Mai H, Lin J, Wen F. Etiology, drug sensitivity profiles and clinical outcome of bloodstream infections: A retrospective study of 784 pediatric patients with hematological and neoplastic diseases. Pediatr Hematol Oncol. 2019 Nov;36(8):482-493.
3. Goel G, Chandy M, Bhattacharyya A, Banerjee S, Chatterjee S, Mullick S, Sinha S, Sengupta K, Dhar K, Bhattacharya S, Rudramurthy S, Chakrabarti A. Mortality associated with candidemia in non-neutropenic cancer patients is not less compared to a neutropenic cohort of cancer patients. Eur J Clin Microbiol Infect Dis. 2017 Dec;36(12):2533-2535.
I suggest to accept the manuscript for publication with minor editing of the discussion, adding the above articles.
Author Response
Replies to reviewer #2
Comments and Suggestions for Authors
The manuscript "Infections During Non-Neutropenic Episodes in Pediatric 2 Cancer Patients – Results from a Prospective Study in Two 3 Major Large European Cancer Centers" provides valuable information to the clinicians about infections occurring in the pediatric ALL/AML/NHL patient population.
The manuscript is well written, and the text/ tables are easy to follow.
We thank the reviewer for these positive comments.
The discussion, including the limitations of the study is OK, but a few other studies/ references could be added to them, such as:
- Chen S, Liu S, Yuan X, Wang H, Wen F. Evaluation of Inflammatory Biomarkers in Pediatric Hematology-Oncology Patients With Bloodstream Infection. J Pediatr Hematol Oncol. 2021 May 1;43(4):e596-e600.
- Chen S, Liu S, Yuan X, Mai H, Lin J, Wen F. Etiology, drug sensitivity profiles and clinical outcome of bloodstream infections: A retrospective study of 784 pediatric patients with hematological and neoplastic diseases. Pediatr Hematol Oncol. 2019 Nov;36(8):482-493.
- Goel G, Chandy M, Bhattacharyya A, Banerjee S, Chatterjee S, Mullick S, Sinha S, Sengupta K, Dhar K, Bhattacharya S, Rudramurthy S, Chakrabarti A. Mortality associated with candidemia in non-neutropenic cancer patients is not less compared to a neutropenic cohort of cancer patients. Eur J Clin Microbiol Infect Dis. 2017 Dec;36(12):2533-2535.
I suggest to accept the manuscript for publication with minor editing of the discussion, adding the above articles.
We thank the reviewer for this excellent suggestion and included all the above mentioned references together with a short statement in the section discussion (references # 13, 16, and 18, lines 133-135, lines 165-167, and lines 172-174)
Reviewer 3 Report
In the present study, Schöning et al, addresses the clinical guidelines for pediatric non-neutropenic cancer patients with fever using broad-spectrum antibiotics, which will enhance the quality of life by decreasing hospitalization. The strength of the study is the sample size and multicenter, however, there are some major limitations in the study:
1) In the introduction, there is no literature on the importance of antibiotics in treating pediatric cancer patients who are neutropenic and non-neutropenic.
2) There is no adequate literature to understand the importance and novelty of the study.
3) In table 1 and 2, there is no p-value to indicate the statistical significance of the results, so it is difficult to draw any conclusions from the table.
4) In methods, there is no description of the sample size calculation that will strengthen the study.
5) There is no data about the survival rates of non-neutropenic and neutropenic cancer patients with different forms of infection; therefore, it is difficult to believe that this study is useful for clinical practice.
Author Response
Replies to reviewer #3
Comments and Suggestions for Authors
In the present study, Schöning et al, addresses the clinical guidelines for pediatric non-neutropenic cancer patients with fever using broad-spectrum antibiotics, which will enhance the quality of life by decreasing hospitalization. The strength of the study is the sample size and multicenter, however, there are some major limitations in the study:
- In the introduction, there is no literature on the importance of antibiotics in treating pediatric cancer patients who are neutropenic and non-neutropenic.
According to the reviewer´s suggestion, we added a statement and a reference that the prompt initiation of intravenous antibiotics that cover both Gram-positive and Gram-negative pathogens has dramatically decreased mortality and has become the standard of care in neutropenic children with cancer (reference # 2, lines 38-43).
In contrast, the importance of antibiotics in the non-neutropenic setting is less clear, which explains the wide variation in the clinical approach ranging from no administration of antibiotics to intravenous combination therapy. According to the reviewer´s suggestion, we included the last sentence in the introduction of the revised version (lines 46-47).
- There is no adequate literature to understand the importance and novelty of the study.
We understand the reviewer’s point. We think that the revised introduction together with the last paragraph in the discussion now address the problem and explain the importance of our data. Please also see our reply to criticism #1 above.
3) In table 1 and 2, there is no p-value to indicate the statistical significance of the results, so it is difficult to draw any conclusions from the table.
The reviewer raises an important point that we had discussed internally prior to the submission of the original manuscript. Both Tables are not easy to read, as there are many columns and lines. For example, in Table 1, columns include all patients, patients with ALL and the respective subgroups SR, MR and HR, AML, NHL, and lines include FUO with neutropenia with and without steroids and without neutropenia with and without steroids, microbiologically documented infections with neutropenia with and without steroids and without neutropenia with and without steroids, and clinically documented infections with neutropenia with and without steroids and without neutropenia with and without steroids, respectively. As the current amount of data in the Tables already limits the readability, we did not include percentages for many of the numbers. Similarly, and even more importantly, we decided not to include the P values in the Tables, which, however are given in the text (e.g. lines 77, 82, 83). By this approach, we provide the reader with all the information that is necessary to draw solid conclusions from the data.
4) In methods, there is no description of the sample size calculation that will strengthen the study.
The study was designed as a prospective observational study over a reasonable time period. We were not aiming to test any hypothesis, nor did we want to show any superiority/inferiority of a certain strategy. Therefore, no sample size calculation was considered necessary..
5) There is no data about the survival rates of non-neutropenic and neutropenic cancer patients with different forms of infection; therefore, it is difficult to believe that this study is useful for clinical practice.
The study was analyzing the characteristics of infections during non-neutropenic episodes, which are by definition of different nature. Unfortunately, many other studies focus on bacteremia in the non-neutropenic patient, but in the clinical setting, it is important to analyze all forms infections in order to institute the best initial treatment (or no treatment).
In the last paragraph of the results section of the original manuscript, we described the outcome of patients with infectious complications occurring during periods of non-neutropenia. Outcome endpoints did not only include death (three patients died of infectious complications, accounting for a mortality of 1.4% of patients with infections during episodes of non-neutropenia), but also severe disease and referral to the ICU were described in the original manuscript. Comparison with survival rates of infectious complications during neutropenic episodes would definitively be interesting, but were not the focus of the current analysis.
Reviewer 4 Report
This is an exciting research paper.
However, I propose a few suggestions to improve the manuscript:
Comment 1: Manuscript layout may be changed (eg Method can be moved up)
Comment 2: Method: Inclusion and exclusion criteria are not clear
Comment 3: Result: what was the final number of patients taken for analysis, 210 0r 203?
Comment 4: : Discussion: First two lines look unnecessary and may be removed or placed somewhere else.
Comment 6:Conclusion; Please mention the limitations.
Author Response
Replies to reviewer #4
This is an exciting research paper.
We thank the reviewer for this positive comment.
However, I propose a few suggestions to improve the manuscript:
Comment 1: Manuscript layout may be changed (eg Method can be moved up)
The layout was done according to the journal´s requirement, and therefore, the section methods was placed at the end.
Comment 2: Method: Inclusion and exclusion criteria are not clear
According to the reviewer´s suggestion we have clarified the inclusion criteria in the section patients and methods, first sentence.
We modified the inclusion criteria and included the statement that all children and adolescents up to 18 years of age who were diagnosed between April 1, 2014 and March 31, 2018, and were treated at the University Children’s Hospitals of Frankfurt or Münster, Germany, for de novo acute lymphoblastic leukemia (ALL), acute myeloid leukemia (AML), relapse of acute leukemia, or non-Hodgkin lymphoma (NHL) could be enrolled in the study. All patients and/or caregivers had to sign informed consent. Patients who did not meet the inclusion criteria were not enrolled (section patients and methods, lines 199-203).
Comment 3: Result: what was the final number of patients taken for analysis, 210 0r 203?
We apologize for the confusion – the final number of analyzed patients was 203, as we excluded the seven patients with leukemia relapse. We clarified this fact in line 64.
Comment 4: : Discussion: First two lines look unnecessary and may be removed or placed somewhere else.
According with the reviewer´s suggestion, we moved the first two lines of the section discussion into the section introduction and modified the start of the discussion (lines 38-43 and 47-47 and lines 125-129).
Comment 6:Conclusion; Please mention the limitations.
According to the reviewer´s suggestion, we included the limitations of our study in the last paragraph of the section discussion (lines 184-186)
Round 2
Reviewer 3 Report
The authors have successfully addressed all my previous comments